# AR4D: Autoregressive 4D Generation from Monocular Videos

## Abstract

Recent advancements in generative models have ignited substantial interest in dynamic 3D content creation (*i.e.*, 4D generation). Existing approaches primarily rely on Score Distillation Sampling (SDS) to infer novel-view videos, typically leading to issues such as limited diversity, spatial-temporal inconsistency and poor prompt alignment, due to the inherent randomness of SDS. To tackle these problems, we propose AR4D, a novel paradigm for SDS-free 4D generation. Specifically, our paradigm consists of three stages. To begin with, for a monocular video that is either generated or captured, we first utilize pre-trained expert models to create a 3D representation of the first frame, which is further fine-tuned to serve as the canonical space. Subsequently, motivated by the fact that videos happen naturally in an autoregressive manner, we propose to generate each frame's 3D representation based on its previous frame's representation, as this autoregressive generation manner can facilitate more accurate geometry and motion estimation. Meanwhile, to prevent overfitting during this process, we introduce a progressive view sampling strategy, utilizing priors from pre-trained large-scale 3D reconstruction models. To avoid appearance drift introduced by autoregressive generation, we further incorporate a refinement stage based on a global deformation field and the geometry of each frame's 3D representation. Extensive experiments have demonstrated that AR4D can achieve state-of-the-art 4D generation without SDS, delivering greater diversity, improved spatial-temporal consistency, better alignment with input prompts and faster generation speed.

## 1 Introduction

In recent years, generative models have made significant strides, allowing for the generation of highly realistic images (Rombach et al., 2022; Zhang et al., 2023; Mou et al., 2024; Podell et al., 2023) and videos (Wu et al., 2023; Blattmann et al., 2023; Villegas et al., 2022; Zhang et al., 2024a) from simple prompts. Building on these successes, numerous studies have sought to extend these capabilities into the domain of dynamic 3D content creation (*i.e.*, 4D generation) (Jiang et al., 2024b; Ren et al., 2023; Zhao et al., 2023; Sun et al., 2024b; Yang et al., 2024a), which is crucial for areas such as virtual reality, gaming, and embodied intelligence.

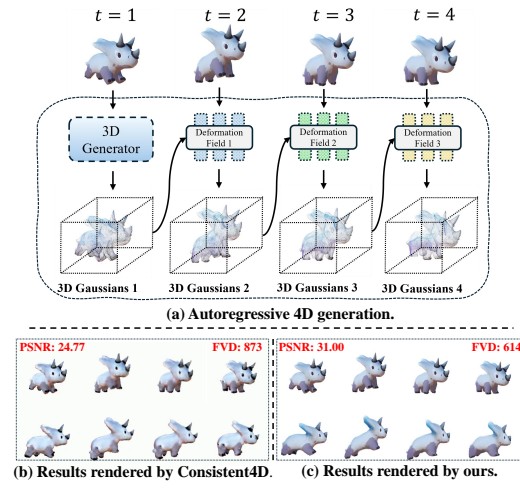

(a) Autoregressive 4D generation.

(b) Results rendered by Consistent4D.    (c) Results rendered by ours.

Figure 1: **Autoregressive 4D generation.**

To achieve this goal, given the lack of large-scale 4D datasets available, existing methods (Jiang et al., 2024b; Ling et al., 2024; Bahmani et al., 2024a; Ren et al., 2023; Zeng et al., 2025; Bahmani et al., 2025; Gao et al., 2024; Miao et al., 2024; Jiang et al., 2024a; Yuan et al., 2024; Li et al., 2024c; Zhao et al., 2023; Zhu et al., 2024b) mainly estimate novel-view videos using Score Distillation Sampling (SDS) (Poole et al., 2022), where knowledge stored in pre-trained multi-modal diffusion models (Liu et al., 2023; 2024a;

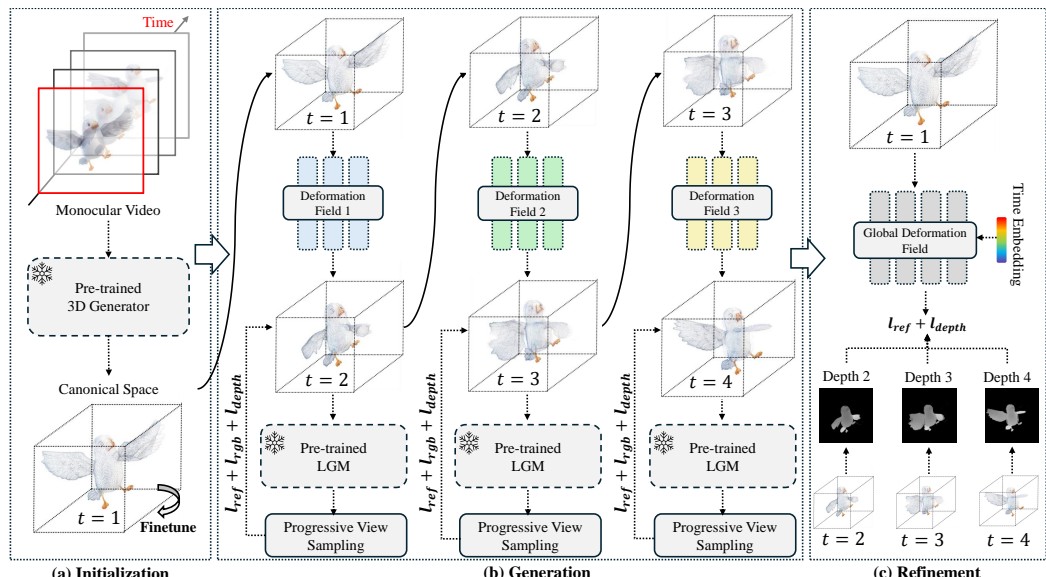

Figure 2: **Paradigm of our proposed AR4D.** To enable SDS-free 4D generation, we propose a three-stage approach consisting of *Initialization*, *Generation*, and *Refinement*. Please see Sec. 4 for more details.

Blattmann et al., 2023) are leveraged to guide the generation process. However, while seemingly reasonable results can be obtained, these SDS-based methods often exhibit several issues (Wang et al., 2024; Liang et al., 2024b; Yi et al., 2023), e.g., limited diversity, spatial-temporal inconsistencies, poor alignment with input prompts, typically resulting in low-quality 4D objects, as demonstrated in Fig. 1(b).

To address these issues, in this paper we propose AR4D, a novel paradigm capable of generating high-quality 4D assets without relying on SDS. Specifically, as shown in Fig. 2, our paradigm is composed of three distinct stages, which are refered to as the *Initialization* stage, the *Generation* stage, and the *Refinement* stage respectively. To begin with, during the *Initialization* stage, as shown in Fig. 1(a), given a monocular video (either generated or captured), we first utilize pre-trained 3D generators (*e.g.*, MVDream (Shi et al., 2023)) to create a 3D representation (*i.e.*, 3D Gaussians (Kerbl et al., 2023)) of the first frame, which is further fine-tuned to serves as the canonical space for the 4D content to be generated.

Subsequently, during the *Generation* stage, to derive the corresponding 4D asset based on the reference video and its first frame's 3D representation without relying on SDS, an intuitive way is to directly employ established 4D reconstruction methods , e.g., Deform 3DGS (Yang et al., 2024b), which learns the deformation of the canonical space through a global deformation field by minimizing the difference between rendered and ground-truth frames. However, unlike typical 4D reconstruction techniques (Wu et al., 2024; Yang et al., 2024b; Li et al., 2024b; Pumarola et al., 2021; Attal et al., 2023) that can utilize multi-view videos or monocular videos with varying viewpoints, our goal relies on monocular videos typically captured from a fixed viewpoint, which poses a greater challenge on accurate motion and geometry estimation, as demonstrated in Fig. 4(a). To address this, motivated by the fact that videos happen naturally in an autoregressive manner, an object's current state in 3D space can be assumed to be transformed from its prior state. To this end, as shown in Fig. 2(b), we propose to generate current frame's 3D representation based on its previous frame's 3D representation, where the dynamics between adjacent frames are represented by an frame-wise local deformation field, rather than a global deformation field for the whole sequence like previous works (Jiang et al., 2024b; Zeng et al., 2025; Ren et al., 2023). Such an autoregressive generation manner facilitates more accurate motion modeling by focusing on localized changes, which is able to better capture subtle, frame-to-frame variations, making the generation process more robust and precise. Moreover, as each timestamp provides only a single fixed-viewpoint frame for supervision, the estimated 3D representation may gradually overfit to this frame over the course of training.

To mitigate this issue, we introduce a progressive view sampling strategy that utilizes priors from pre-trained large-scale 3D reconstruction models (e.g., LGM (Tang et al., 2024)) to progressively provide pseudo views as additional supervisions, which we find can guarantee the spatial-temporal consistency of the underlying geometry to a large extent.

After obtaining each frame's 3D representation, it is observed that due to accumulated errors introduced by autoregressive generation, the 3D representations of later frames exhibit noticeable appearance drift, which affects the quality of the generated results, as demonstrated in Fig. 5(a). To address this issue, as shown in Fig. 2(c), we further propose a ***Refinement*** stage, based on the observation that the geometric structure of each frame remains relatively stable (Niemeyer et al., 2022). Therefore, we take the 3D representation of the first frame as the canonical space and construct a global deformation field. This field is constrained by the geometric structures of different frames, ensuring that the deformations of the canonical space are kept in check. By doing so, we can significantly reduce appearance drift and guarantee spatial-temporal consistency in the generated 4D assets.

Our main contributions can be summarized as follows:

- We propose AR4D, a novel paradigm for generating high-quality 4D assets from monocular videos, bypassing the limitations of Score Distillation Sampling (SDS).
- We propose to generate each frame's 3D representation autoregressively using a local deformation field. This process is further improved through a progressive view sampling strategy, enabling precise geometry and motion estimation.
- To mitigate the issue of accumulated errors, we propose a refinement stage based on a global deformation field and the extracted geometry of each frame's 3D representation, ensuring the spatial-temporal consistency of generated 4D contents.
- Extensive experiments have demonstrated that our proposed AR4D can achieve state-of-the-art performance without SDS, with greater diversity, improved spatial-temporal consistency, better alignment with input prompts and faster generation speed.

## 2 RELATED WORKS: 4D GENERATION

Prior-based approaches enable 4D generation by either training a generalized model through large-scale multi-modal datasets (Deitke et al., 2023) or integrating pre-trained models directly. For example, methods such as (Xie et al., 2024; Li et al., 2024a; Liang et al., 2024a; Zhang et al., 2024b) proposed to generate multi-view videos by training a multi-view video diffusion model, which are subsequently processed with 4D reconstruction techniques to produce corresponding 4D assets. To expedite the generation process, L4GM (Ren et al., 2024) introduced the first 4D Large Reconstruction Model capable of producing animated objects in a single feed-forward pass within just one second. Recently, inspired by the powers of video generative models, several approaches (He et al., 2024b; Bahmani et al., 2024b; Yu et al., 2024; Xu et al., 2024; Hou et al., 2024) have endowed them with camera control capabilities, allowing for generating videos with varying viewpoints. While photorealistic 4D contents can be achieved, these methods often incur high pre-training costs, and the pre-trained scenes may not be well-suited to the target scene. Another category of 4D generation methods adopted a scene-specific optimization approach to produce better 4D contents tailored to each individual scene. To achieve this, mainstream methods (Jiang et al., 2024b; Ling et al., 2024; Bahmani et al., 2024a; Ren et al., 2023; Zeng et al., 2025; Bahmani et al., 2025; Gao et al., 2024; Miao et al., 2024; Jiang et al., 2024a; Yuan et al., 2024; Li et al., 2024c; Zhao et al., 2023; Zhu et al., 2024b) primarily distilled knowledge from pre-trained multimodal models (i.e., SDS) to guide the generation process. For instance, Consistent4D (Jiang et al., 2024b) achieved Video-to-4D generation by combining SDS with dynamic NeRF (Mildenhall et al., 2021), followed by a video enhancer to produce high-quality 4D objects. Addressing NeRF's limitations, DreamGaussian4D (Ren et al., 2023) introduced the 3DGS (Kerbl et al., 2023) representation, enhanced with texture refinement for fast 4D generation. Recently, STAG4D (Zeng et al., 2025) proposed an innovative approach that can generate anchor multi-view sequences, followed by 4D Gaussian field fitting using SDS to improve 4D generation quality. While these SDS-based methods can achieve reasonable results, they are often hindered by issues (Wang et al., 2024; Liang et al., 2024b; Yi et al., 2023) such as limited diversity, spatial-temporal inconsistency, and poor alignment with input prompts, significantly limiting their practical applications. In contrast, in this paper we propose AR4D, a novel paradigm that is SDS-free for better 4D generation.

## 3 PRELIMINARIES: 3DGS

3D Gaussian Splatting (3DGS) (Kerbl et al., 2023) has shown impressive capability in novel view synthesis, enabling photorealistic novel views to be rendered in real-time. Different from NeRF (Mildenhall et al., 2021) that encodes scene properties into neural networks, 3DGS (denoted by $\mathbf{G}$) leverages millions of anisotropic ellipsoids to capture scene geometry and appearance, with each ellipsoid (i.e., 3D Gaussian) parameterized by position $\mu \in \mathbb{R}^3$, opacity $\alpha \in \mathbb{R}$, covariance $\mathbf{\Sigma} \in \mathbb{R}^{3 \times 3}$ (calculated from scale $\mathbf{s} \in \mathbb{R}^3$ and rotation $\mathbf{r} \in \mathbb{R}^3$), and color $\mathbf{c} \in \mathbb{R}^3$. For simplicity, in this paper we represent the attributes of all ellipsoids collectively as $\mathbf{G} = \{\mu, \alpha, \mathbf{s}, \mathbf{r}, \mathbf{c}\}$.

## 4 METHODS

For a monocular video $V = \{v_1, v_2, \ldots, v_F\}$ (either generated or captured from a fixed viewpoint) with $F$ frames, our objective is to generate its corresponding 4D content without relying on SDS, while enhancing diversity, spatial-temporal consistency, and alignment with the input prompts.

### 4.1 INITIALIZATION

In the first stage, we aim to obtain a 3D representation, which is used to serve as the canonical space for its 4D counterpart. Leveraging recent advances in 3D generation, we first employ a pre-trained multi-view diffusion model to generate several novel views of the first frame, followed by a pre-trained large-scale 3D reconstruction model to recover the corresponding 3D representation (i.e., 3D Gaussians $\mathbf{G}_1^{init} = \{\mu_1^{init}, \alpha_1^{init}, \mathbf{s}_1^{init}, \mathbf{r}_1^{init}, \mathbf{c}_1^{init}\}$) from these generated views. However, as shown in Fig. 3(b), due to the inherent limitations of these pre-trained models, the generated 3D

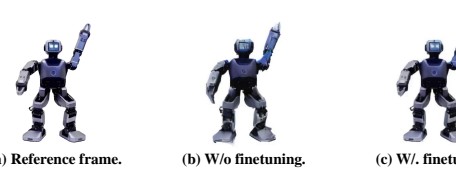

(a) Reference frame.    (b) W/o finetuning.    (c) W/. finetuning.

Figure 3: Ablation studies on finetuning the 3D Gaussians in the *Initialization* stage reveal that finetuning can capture finer texture details in the reference frame, enhancing the quality of subsequent generation.

Gaussians often fail to accurately capture the fine-grained texture details of the reference frame $v_1$, presenting additional challenges for the subsequent reconstruction stage.

To mitigate this issue, we propose a simple yet effective method to fine-tune the obtained 3D Gaussians. Specifically, we keep the parameters $\{\alpha_1^{init}, \mathbf{s}_1^{init}, \mathbf{r}_1^{init}\}$ that influence each gaussian's geometry unchanged, while only optimizing $\{\mu_1^{init}, \mathbf{c}_1^{init}\}$ to ensure consistency in rendering with the reference frame $v_1$ without harming the overall geometry, using the following equation:

$$\mu_1^{ft}, \mathbf{c}_1^{ft} = \underset{\mu_1^{init}, \mathbf{c}_1^{init}}{\arg\min} \|R^{ref}(\mathbf{G}_1^{init}) - v_1\|_2, \tag{1}$$

where $R^{ref}$ means rendering $\mathbf{G}_1^{init}$ at the view of the reference frame, and the fine-tuned $\mathbf{G}_1$ is thus formulated as $\mathbf{G}_1 = \{\mu_1^{ft}, \alpha_1^{init}, \mathbf{s}_1^{init}, \mathbf{r}_1^{init}, \mathbf{c}_1^{ft}\}$.

As shown in Fig. 3(c), the fine-tuned 3D Gaussians can produce results that are better aligned with the reference frame, thereby facilitating the subsequent generation process.

### 4.2 GENERATION

**Autoregressive generation.** To generate the 3D Gaussians for each frame based on $V$ and $\mathbf{G}_1$, a straightforward way is to directly apply common 4D reconstruction methods (e.g., Deform 3DGS (Yang et al., 2024b)), where $\mathbf{G}_1$ serves as the canonical space, and a global deformation field $F_\theta$ is used to estimate the motion of $\mathbf{G}_1$ at different timestamps by minimizing the difference between the rendered videos and $V$. However, unlike typical 4D reconstruction tasks that can leverage multi-view videos or monocular videos with varying viewpoints, we only have access to monocular videos with a fixed viewpoint, which creates additional challenges for accurate geometry and motion estimation, often resulting in severe artifacts, as demonstrated in Fig. 4(a).

To address this problem, we propose to leverage the autoregressive nature of videos, which indicates that the 3D Gaussians of consecutive frames undergo only minor deformations. As a result, the 3D Gaussians of the current frame can be seen as being heavily influenced by those of its previous frame. Based on this motivation, we propose to perform the 4D generation from $V$ and $\mathbf{G}_1$ in an autoregressive manner.

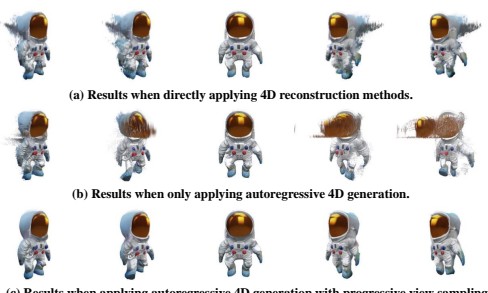

(a) Results when directly applying 4D reconstruction methods.

(b) Results when only applying autoregressive 4D generation.

(c) Results when applying autoregressive 4D generation with progressive view sampling.

Figure 4: Ablation studies on finetuning the 3D Gaussians in the *Initialization* stage reveal that finetuning can capture finer texture details in the reference frame, enhancing the quality of subsequent generation.

Specifically, as shown in Fig. 2(b), for each pair of adjacent frames $v_i$ and $v_{i+1}$, we utilize an independent MLP-based local deformation field $F_{\theta_i}$ to model the deformations between their corresponding 3D Gaussians $\mathbf{G}_i = \{\mu_i, \alpha_i, \mathbf{s}_i, \mathbf{r}_i, \mathbf{c}_i\}$ and $\mathbf{G}_{i+1} = \{\mu_{i+1}, \alpha_{i+1}, \mathbf{s}_{i+1}, \mathbf{r}_{i+1}, \mathbf{c}_{i+1}\}$, which is formulated as follows:

$$\{\delta_{\mu_i}, \delta_{\alpha_i}, \delta_{\mathbf{s}_i}\} = F_{\theta_i}(\gamma(\mu_i)), \quad \begin{cases} \mu_{i+1} = \mu_i + \delta_{\mu_i} \\ \alpha_{i+1} = \alpha_i + \delta_{\alpha_i} \\ \mathbf{s}_{i+1} = \mathbf{s}_i + \delta_{\mathbf{s}_i} \\ \mathbf{r}_{i+1} = \mathbf{r}_i \\ \mathbf{c}_{i+1} = \mathbf{c}_i \end{cases}, \tag{2}$$

where $\gamma$ is the positional encoding operation that is denoted as follows:

$$\gamma(\boldsymbol{x}) = (\sin(2^0\boldsymbol{x}), \cos(2^0\boldsymbol{x}), \cdots, \sin(2^{L-1}\boldsymbol{x}), \cos(2^{L-1}\boldsymbol{x})), \tag{3}$$

where $L$ is a hyperparameter that is usually set to 10.

To obtain $\mathbf{G}_{i+1}$ based on $\mathbf{G}_i$, we minimize the difference between the rendered frame $\hat{v}_{i+1} = R^{\text{ref}}(\mathbf{G}_{i+1})$ and the reference frame $v_{i+1}$, as expressed by:

$$\{\theta_i, \mu_i, \alpha_i, \mathbf{s}_i, \mathbf{r}_i, \mathbf{c}_i\} = \underset{\{\theta_i, \mu_i, \alpha_i, \mathbf{s}_i, \mathbf{r}_i, \mathbf{c}_i\}}{\operatorname{argmin}} l_{ref},$$

$$l_{ref} = \lambda\|\hat{v}_{i+1} - v_{i+1}\|_1 + (1-\lambda)\text{SSIM}(\hat{v}_{i+1}, v_{i+1}) \tag{4}$$

where $R^{\text{ref}}$ means rendering $\mathbf{G}_{i+1}$ at the view of $v_{i+1}$, SSIM means the loss function used to measure the SSIM metric between $\hat{v}_{i+1}$ and $v_{i+1}$, $\lambda$ is a balancing parameter which is set to 0.8.

**Progressive view sampling strategy.** As demonstrated in Fig. 4(b), during the process of autoregressive generation, since each timestamp provides only a single fixed-viewpoint frame for supervision, the generated 3D Gaussians tend to overfit to the reference frames, particularly for the later frames in $V$, leading to significant artifacts in novel views.

To solve this problem, we propose to leverage the powers of pre-trained large-scale 3D reconstruction models (Tang et al., 2024) by introducing pseudo novel views as additional supervisions. To achieve this, the major challenge lies on how to obtain appropriate novel views that not only prevent overfitting but also reliable enough to ensure accurate and spatial-temporal-consistent generation.

To this end, we propose a simple yet effective progressive view sampling strategy. Specifically, during the generation process of $\mathbf{G}_{i+1}$, we first render several orthogonal views (including the reference view) of $\mathbf{G}_{i+1}$, which are then fed into the large-scale 3D reconstruction model to create a pseudo 3D Gaussians $\hat{\mathbf{G}}_{i+1}$. Subsequently, considering that during the early stages of optimizing, views rendered by $\hat{\mathbf{G}}_{i+1}$, especially those close to the reference view, are highly reliable, we initially constrain $\mathbf{G}_{i+1}$ by randomly sampling novel views within this close view range using $\hat{\mathbf{G}}_{i+1}$ as additional supervision. With training in progress, the range of sampled viewpoints is progressively expanded to prevent overfitting.

As a result, the progressive view sampling strategy is denoted as follows:

$$N_u = \min(N_{\max}, \lfloor u/\eta \rfloor + N_{start}), \tag{5}$$

where $N_u$ represents the maximum azimuth angle that can be sampled at the $u$-th iteration, $N_{\max}$ is the upper limit of $N_u$, $N_{start}$ is the initial azimuth sampling limit when reconstructing $\mathbf{G}_{i+1}$, and $\eta$ is a hyperparameter controlling the rate at which $Nu$ increases. During the sampling process, the elevation angle and radius are kept the same as the reference view.

Based on this strategy, for a sampled novel view $N_{samp} \sim \mathcal{U}(-N_u, N_u)$, $\mathbf{G}_{i+1}$ is further regularized with the following equations:

$$\{\theta_i, \mu_i, \alpha_i, \mathbf{s}_i, \mathbf{r}_i, \mathbf{c}_i\} = \underset{\{\theta_i, \mu_i, \alpha_i, \mathbf{s}_i, \mathbf{r}_i, \mathbf{c}_i\}}{\arg\min} \, l_{rgb} + l_{depth}, \tag{6}$$

where

$$
\begin{aligned}
l_{rgb} &= \|R^{N_{samp}}(\mathbf{G}_{i+1}) - R^{N_{samp}}(\hat{\mathbf{G}}_{i+1})\|_1 \\
l_{depth} &= \|R^{N_{samp}}_{depth}(\mathbf{G}_{i+1}) - R^{N_{samp}}_{depth}(\hat{\mathbf{G}}_{i+1})\|_1,
\end{aligned} \tag{7}
$$

with $R^{N_{samp}}$ denoting the rendering of images of $\mathbf{G}_{i+1}$ and $\hat{\mathbf{G}}_{i+1}$ at view $N_{samp}$, and $R^{N_{samp}}_{depth}$ representing the rendering of their corresponding depth maps at view $N_{samp}$.

As demonstrated in Fig. 4(c), the proposed autoregressive generation combined with the progressive view sampling strategy enables accurate motion and geometry estimation significantly.

## 4.3 REFINEMENT

As shown in Fig. 5(a), performing 4D generation in an autoregressive manner introduces accumulated errors, resulting in noticeable appearance drift, particularly in the later frames of the monocular video $V$.

To address this issue, we propose a refinement stage motivated by the observation that while high-frequency appearance may drift, the geometry (e.g., depth map) of each frame remains relatively low-frequency (Niemeyer et al., 2022) and stable throughout training, as demonstrated in Fig. 5. As a result, in this stage, $\mathbf{G}_1 = \{\mu_1, \alpha_1, \mathbf{s}_1, \mathbf{r}_1, \mathbf{c}_1\}$ is treated as the canonical space, and a global deformation field $F_\theta$, constrained by each frame's depth map, is used to model the deformations across frames, resulting in $\{\mathbf{G}^{re}_k = \{\mu^{re}_k, \alpha^{re}_k, \mathbf{s}^{re}_k, \mathbf{r}^{re}_k, \mathbf{c}^{re}_k\}\}^F_{k=2}$.

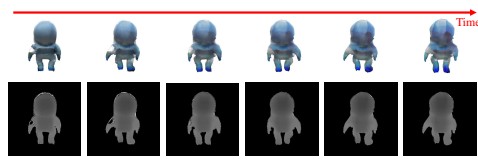 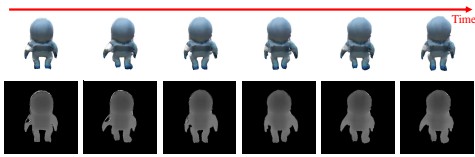

(a) Appearance drift caused by autoregressive generation.

(b) With the refinement stage, no obvious appearance drift is observed.

Figure 5: Results of the **_Refinement_** stage demonstrate its effectiveness in addressing appearance drift. While appearance may fluctuate, the geometry (evident in the consistent depth map) remains stable, enabling the generation of spatial-temporal consistent 4D contents.

Specifically, the relationship between $\mathbf{G}_1$ and $\mathbf{G}^{re}_k$ is formulated as follows:

$$\{\delta^{re}_{\mu_k}, \delta^{re}_{\alpha_k}, \delta^{re}_{\mathbf{s}_k}\} = F_\theta(\gamma(\mu_1), k),
\begin{cases}
\mu^{re}_k = \mu_1 + \delta^{re}_{\mu_k} \\
\alpha^{re}_k = \alpha_1 + \delta^{re}_{\alpha_k} \\
\mathbf{s}^{re}_k = \mathbf{s}_1 + \delta^{re}_{\mathbf{s}_k} \\
\mathbf{r}^{re}_k = \mathbf{r}_1 \\
\mathbf{c}^{re}_k = \mathbf{c}_1
\end{cases}, \tag{8}$$

with $\mathbf{G}_1$ and $F_\theta$ optimized using the following equation:

$$\{\theta, \mu_1, \alpha_1, \mathbf{s}_1, \mathbf{r}_1, \mathbf{c}_1\} = \underset{\{\theta, \mu_1, \alpha_1, \mathbf{s}_1, \mathbf{r}_1, \mathbf{c}_1\}}{\arg\min} \, l^{re}_{ref} + l^{re}_{depth}, \tag{9}$$

where

$$
\begin{aligned}
l^{re}_{ref} &= \mathbb{E}_k[\|R^{ref}(\mathbf{G}_k) - R^{ref}(\mathbf{G}^{re}_k)\|_1] \\
l^{re}_{depth} &= \mathbb{E}_k[\|R^{N^{re}_{samp}}_{depth}(\mathbf{G}_k) - R^{N^{re}_{samp}}_{depth}(\mathbf{G}^{re}_k)\|_1],
\end{aligned} \tag{10}
$$

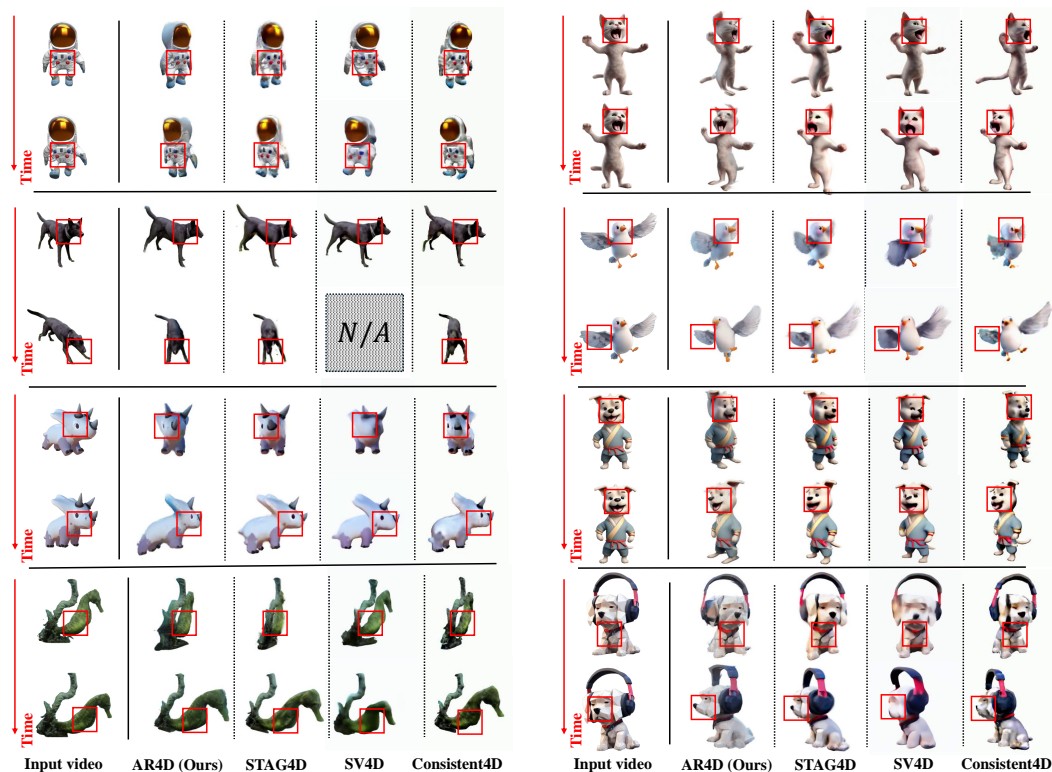

(a) Qualitative comparisons on Video-to-4D.          (b) Qualitative comparisons on Text-to-4D.

Figure 6: Qualitative comparisons of our proposed AR4D with other state-of-the-art methods. Our method generates more detailed results with improved alignment to input prompts. *N/A* indicates that the corresponding method fails to generate novel views for the current frame.

$\mathbf{G}_k$ denotes the 3D Gaussians obtained during the reconstruction stage for the $k$-th frame, $R^{ref}$ denotes rendering of $\mathbf{G}_k$ and $\mathbf{G}_k^{re}$ at view of the reference frame $v_k$, $N_{samp}^{re}$ refers to a randomly sampled viewpoint within the view space, and $R_{depth}^{N_{samp}^{re}}$ represents the rendering of the depth maps of $\mathbf{G}_k$ and $\mathbf{G}_k^{re}$ from the viewpoint $N_{samp}^{re}$.

As demonstrated in Fig. 5(b), this refinement ensures that each frame's geometry, obtained in the *Generation* stage, remains unchanged while its appearance is directly deformed from the same 3D Gaussians $\mathbf{G}_1$, preventing significant appearance drift and thus improving spatial-temporal consistency.

## 5 EXPERIMENTS

**Datasets and metrics.** Following the experimental protocols outlined by STAG4D (Zeng et al., 2025), we use the provided datasets to conduct experiments. Specifically, our experiments cover both video-to-4D and text-to-4D generation tasks across approximately **50 diverse scenes** (where previous methods such as Consistent4D contains only 8 scenes). The image-to-4D task is performed in two steps: first, converting the image to video, followed by the video-to-4D transformation. To evaluate the quality of the generated results, we report PSNR, SSIM (Wang et al., 2004), and LPIPS (Zhang et al., 2018) to assess the alignment between the rendered videos and the ground truth. Additionally, we report CLIP similarity (Radford et al., 2021) and FVD scores (Unterthiner et al., 2018) to measure the consistency between the rendered novel views and the reference views. For the PSNR and SSIM evaluation, we selected three viewpoints: the reference view, as well as two novel views obtained by rotating the reference view ±30° along the azimuth, with the elevation fixed. The reported result is the average across these three views. We believe that including the reference view is important,

| Method | Consistent4D | SV4D | STAG4D | Ours |
|---|---|---|---|---|
| PSNR↑ | 24.77 | 28.23 | 29.91 | **31.00** |
| SSIM↑ | 0.91 | 0.92 | 0.94 | **0.97** |
| LPIPS↓ | 0.11 | 0.06 | 0.05 | **0.02** |
| CLIP-S↑ | 0.90 | 0.89 | 0.90 | **0.92** |
| FVD↓ | 873 | - | 737 | **617** |
| FVD-16↓ | 611 | 592 | 573 | **478** |

Table 2: Quantitative comparisons of our method with other state-of-the-art methods on Video-to-4D. The best, second-best, and third-best entries are marked in red, orange, and yellow.

| Method | Consistent4D | SV4D | STAG4D | Ours |
|---|---|---|---|---|
| PSNR↑ | 23.38 | 28.34 | 30.26 | **31.06** |
| SSIM↑ | 0.88 | 0.93 | 0.95 | **0.98** |
| LPIPS↓ | 0.17 | 0.09 | 0.06 | **0.03** |
| CLIP-S↑ | 0.89 | 0.88 | 0.90 | **0.92** |
| FVD↓ | 1250 | - | 1065 | **890** |
| FVD-16↓ | 1084 | 1032 | 947 | **681** |

Table 3: Quantitative comparisons of our method with other state-of-the-art methods on Text-to-4D. The best, second-best, and third-best entries are marked in red, orange, and yellow.

as it reflects how well the generated 4D content preserves fidelity to the input monocular video, which is essential for ensuring spatial and temporal consistency. We intentionally did not include views from the back side of the object (*i.e.*, 180° azimuthal rotation) because such views are typically fully hallucinated based on the reference frame, and no method—ours or others—can be perfectly consistent with the ground truth in those regions. Including them in PSNR computation would disproportionately

| Method | LPIPS↓ | PSNR↑ | CLIP↑ | FVD↓ |
|---|---|---|---|---|
| L4GM | 0.124 | 27.85 | 0.94 | 734 |
| Consistent4D | 0.160 | 25.68 | 0.87 | 1160 |
| STAG4D | 0.126 | 27.99 | 0.91 | 1013 |
| DG4D | 0.167 | 25.32 | 0.87 | 1224 |
| 4DGen | 0.136 | 27.42 | 0.90 | 998 |
| SV4D | 0.118 | 27.65 | – | – |
| **Ours** | **0.096** | **29.34** | **0.95** | **675** |

Table 1: Quantitative comparisons on the Consistent4D (Jiang et al., 2024b) benchmark.

penalize all methods and fail to reflect meaningful performance differences. For the LPIPS evaluation, since it measures perceptual similarity using features extracted from pre-trained networks, we chose four more diverse viewpoints to better capture appearance similarity across varying viewpoints. Specifically, we evaluated LPIPS on views rotated by $-15°$, $75°$, $165°$, and $255°$ from the reference view (azimuth), and reported the average value. This setup reflects a broader perceptual assessment of the generated results. We computed FVD and CLIP score in the same way. We also present comparisons on the Consistent4D (Jiang et al., 2024b) benchmark with **8 scenes** to further highlight the superiority of our method. Kindly refer to supplementary materials for more details.

**Baselines.** On the STAG4D dataset (Zeng et al., 2025) with about 50 diverse scenes, we compare our proposed AR4D with several state-of-the-art methods, including Consistent4D (Jiang et al., 2024b), SV4D (Xie et al., 2024), and STAG4D (Zeng et al., 2025). Furthermore, on the Consistent4D benchmark (Jiang et al., 2024b) containing 8 scenes, we compare AR4D with a broader set of methods, including L4GM (Ren et al., 2024), Consistent4D, STAG4D, DG4D (Ren et al., 2023), 4DGen, and SV4D.

### 5.1 COMPARISONS WITH STATE-OF-THE-ART METHODS

**Comparisons on the STAG4D dataset (Zeng et al., 2025).** As shown in Fig. 6, given a monocular video, Consistent4D produces over-saturated outputs with a blurred appearance, limited by the intrinsic constraints of SDS. Similarly, although STAG4D can reduce over-saturation to some degree, the results still exhibit noticeable noise and unrealistic, fabricated patterns. For SV4D, as a general 4D generative model, the domain gap issue leads to highly blurred novel views, restricting it to processing short input videos of only 21 frames. In contrast, our proposed AR4D can achieve clearer results with enhanced alignment to input videos and improved spatial-temporal consistency. We provide more visualizations in the supplementary materials. As demonstrated in Tab. 2 and Tab. 3, our proposed method can achieve the highest performance, with an average improvement of 1 dB in PSNR, demonstrating that AR4D can generate 4D assets closely aligned with the input. Moreover, we can also achieve the best CLIP similarity and FVD-score, indicating superior spatial-temporal consistency in the generated 4D objects.

**Comparisons on the Consistent4D dataset (Jiang et al., 2024b).** As demonstrated in Fig. 7 and Tab. 1, SDS-based approaches generally yield blurry generations due to inherent limitations of the SDS formulation, leading to lower CLIP similarity and higher FVD scores. While L4GM delivers superior visual fidelity, it, like SV4D, is hindered by domain gap issues and exhibits limited generalization to unseen scenes outside the training distribution. In contrast, our method attains the best overall performance by leveraging strong priors from expert models and its SDS-free design, thereby enhancing both generalization capability and spatio-temporal consistency.

**(a) Reference Image** **(b) Consistent4D** **(c) STAG4D** **(d) Ours**

Figure 7: Qualitative comparisons on the Consistent4D (Jiang et al., 2024b) benchmark.

**Generation efficiency and computation cost.** Thanks to the SDS-free nature of our method, the optimization for each frame's 3D Gaussian scene **takes approximately 30–40 seconds on a single A100 GPU**, resulting in a total generation time of about 15–20 minutes for a typical 30-frame video, with a peak GPU memory consumption of 30–40 GB VRAM. This is **significantly faster than prior state-of-the-art 4D generation methods** under similar hardware conditions. For instance, Consistent4D (Jiang et al., 2024b) requires approximately 1.5–2 hours per video, STAG4D (Zeng et al., 2025) around 1 hour, and 4DGen also about 1 hour. These comparisons highlight the computational efficiency and practicality of our method.

## 5.2 ABLATION STUDIES

| Init-ft | ✓ | ✗ | ✓ | ✓ | ✓ | ✓ |
|---|---|---|---|---|---|---|
| AR | ✓ | ✓ | ✓ | ✓ | ✗ | ✓ |
| PVS | ✓ | ✓ | ✓ | ✗ | ✓ | ✗ |
| Refine | ✓ | ✓ | ✗ | ✓ | ✓ | ✗ |
| PSNR↑ | **31.00** | 30.43 | 30.24 | 30.86 | 30.53 | 30.74 |
| SSIM↑ | **0.97** | 0.95 | 0.95 | 0.96 | 0.94 | 0.95 |
| LPIPS↓ | **0.02** | 0.04 | 0.08 | 0.04 | 0.10 | 0.10 |
| FVD↓ | **617** | 681 | 712 | 1532 | 1026 | 1637 |

Table 4: Ablation studies on the Video-to-4D dataset, where **Init-ft** means finetuning the 3D Gaussians obtained in the *Initialization* stage, **AR** and **PVS** means autoregressive generation and progressive view sampling strategy, **Refine** means whether incorporating the *Refinement* stage.

**(a) Input monocular video.**

**(b) Results rendered without autoregressive generation.**

**(c) Results rendered with autoregressive generation.**

Figure 8: Ablation study on the effect of autoregressive generation: results show that incorporating autoregressive modeling significantly enhances both motion continuity and geometric consistency, resulting in more realistic results.

To showcase the effectiveness of our design choices, we conduct both quantitative and qualitative ablation studies on the task of video-to-4D. As shown in Tab. 4 and Fig. 3, when omitting finetuning the 3D Gaussians obtained in the *Initialization* stage, a performance drop is observed due to the inherent limitations of adopted pre-trained 3D generators. Similarly, when removing the refinement stage, both alignment with input videos and spatial-temporal consistency are negatively influenced, owning to the appearance drift mentioned in Sec. 4.3 and Fig. 5. As demonstrated in Fig. 4, if we remove the progressive view sampling strategy, the generated 4D assets overfit to input videos, resulting in relatively high reconstruction metrics (e.g., PSNR) but significantly lower FVD scores. Additionally, as demonstrated in Fig. 8, if we remove the autoregressive generation, the performance also drops due to the lack of precise motion and geometry estimation. More visualizations are provided in the supplementary materials.

## 6 CONCLUSION

In this paper, we introduce AR4D, a novel approach for SDS-free 4D generation from monocular videos. AR4D operates in three stages: *1) Initialization:* Pre-trained 3D generators are employed to extract 3D Gaussians from the video's first frame, which are then fine-tuned to establish the canonical space for its 4D counterpart. *2) Generation:* For more accurate motion and geometry estimation, 3D Gaussians are generated for each frame in an autoregressive manner, complemented by a progressive view sampling strategy to mitigate overfitting. *3) Refinement:* To counteract appearance drift introduced by autoregressive generation, a global deformation field works in conjunction with per-frame geometry to achieve detailed refinement. Experiments have demonstrated that our method can achieve state-of-the-art 4D generation, with greater diversity, improved spatial-temporal consistency, and better alignment with input prompts.

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

## A    EXPERIMENTAL DETAILS

### A.1    IMPLEMENTATION DETAILS

For 4D object generation, in the ***Initialization*** stage, we first use MVDream (Shi et al., 2023) to generate four orthogonal views of the first frame from the input video. These views are then fed into LGM (Tang et al., 2024) to obtain the corresponding 3D Gaussian representations. Due to the inherent limitations of MVDream, the generated novel views may not always meet quality expectations; in such cases, multiple attempts are encouraged to achieve the most satisfactory results for subsequent stages. After obtaining the 3D representation for the first frame, we fine-tune these 3D Gaussians to better align with the first frame itself. This fine-tuning is performed with a learning rate of $1 \times 10^{-5}$ over 1000 iterations. During the ***Generation*** stage, the input video is assumed to be bind with a camera pose of azimuth angle equals to $0°$, elevation angle equals to $0°$, and radius equals to 1.5. To achieve progressive view sampling, we first render four orthogonal views of the 3D representation that is currently being optimized, with azimuth angle equals to $\{0°, 90°, 180°, 270°\}$ respectively, and elevation angle equals to $0°$, radius equals to 1.5. These views are then input into LGM to generate additional pseudo-labels, on the purpose of prevent overfitting. During the ***Refinement*** stage, the MLP-based global deformation field may occasionally converge to a local optimum, causing training collapse. In such cases, we recommend re-initializing the network or using an improved architecture, such as the one proposed by (Zhu et al., 2024a). All results are rendered at a resolution of $512 \times 512$, which is the maximum resolution supported by LGM for processing.

## B    MORE RELATED WORKS

### B.1    3D GENERATION

The rapid advancements in image generation (Rombach et al., 2022; Zhang et al., 2023; Mou et al., 2024; Podell et al., 2023) and video generation (Wu et al., 2023; Blattmann et al., 2023; Villegas et al., 2022; Zhang et al., 2024a) have sparked significant interest in the field of 3D generation. To address the challenge of limited 3D datasets, Dreamfusion (Poole et al., 2022) proposed the concept of SDS, which has inspired numerous follow-up works (Lin et al., 2023; Chen et al., 2023; Qian et al., 2023; Liu et al., 2024b; Hu et al., 2024). To overcome the inherent limitations of SDS, various improvements have been proposed. For instance, ProlificDreamer (Wang et al., 2024) proposed VSD for synthesizing objects with higher diversity. DreamTime (Huang et al., 2023) proposed a timestep annealing strategy to overcome the over-saturation problem of SDS. Moreover, LucidDreamer (Liang et al., 2024b) introduced interval score sampling for high-fidelity generation. DreamGaussian (Tang et al., 2023) introduced the 3D Gaussian Splatting (3DGS) representation, enabling significantly faster 3D generation, where realistic 3D objects can be synthesized within minutes. Recently, with the development of large-scale 3D datasets (Deitke et al., 2023), several methods (Liu et al., 2023; 2024a; Shi et al., 2023; Tang et al., 2024; Hong et al., 2023) have explored building generalized frameworks for 3D generation, where diverse 3D contents can be generated in a feed-forward process without per-scene optimization. In this paper, we aim to extend the capabilities of existing 3D generation models to the task of 4D generation, without relying on SDS.

### B.2    4D RECONSTRUCTION

4D reconstruction (*i.e.*, dynamic 3D reconstruction) has long been a challenging problem in computer vision and graphics, attracting growing attention in recent years. Early approaches (Pumarola et al., 2021; Attal et al., 2023; Fridovich-Keil et al., 2023; Wang et al., 2023) extended the static NeRF (Mildenhall et al., 2021) framework to dynamic scenes, achieving photorealistic results but suffering from extremely slow training and rendering speeds. Recently, inspired by the powerful

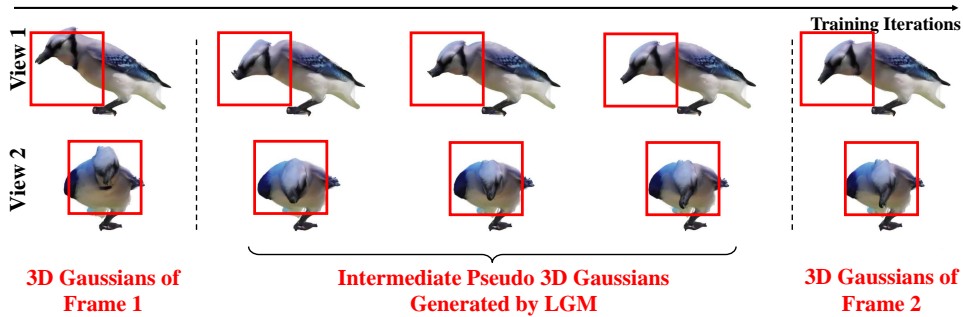

Figure 9: Pseudo 3D Gaussians generated by LGM.

abilities of 3DGS (Kerbl et al., 2023), researchers have begun to explore its integration into 4D reconstruction to improve efficiency. To achieve this goal, similar to (Pumarola et al., 2021), mainstream methods (Wu et al., 2024; Yang et al., 2024b; Pumarola et al., 2021; Attal et al., 2023) typically leverage a canonical space paired with a global deformation field to model motions across frames. More recently, several methods (Sun et al., 2024a; Luiten et al., 2024; He et al., 2024a) proposed to realize efficient 4D reconstruction on a per-frame training manner from multi-view videos, either by introducing Neural Transformation Cache or additional priors such as optical flows. In contrast, our approach targets 4D generation from monocular videos with a fixed viewpoint, a significantly more challenging task that demands precise estimation of motion, geometry, and appearance.

## C    PSEUDO 3D GAUSSIANS GENERATED BETWEEN ADJACENT FRAMES.

As shown in Fig. 9, we provide pseudo 3D Gaussians between adjacent frames during the autoregressive generation process, where LGM ensures reasonable orthogonal results for supervision.

## D    MORE VISUALIZATIONS OF ABLATION STUDIES

To demonstrate the effectiveness of our design choices, we provide additional visualizations of the generated multi-view videos from the ablation studies conducted in Sec. 5.3. As shown in Fig. 10(a), directly applying typical 4D reconstruction methods results in noticeable artifacts due to the use of monocular videos with a fixed viewpoint for supervision, rather than multi-view videos or monocular videos with varying viewpoints. When relying solely on autoregressive generation, severe artifacts tend to appear, especially in later frames, due to the overfitting problem, as shown in Fig. 10(b). Similarly, as shown in Fig. 11(b), removing autoregressive generation (*i.e.*, using only the progressive view sampling strategy) makes accurate motion estimation difficult, particularly in frames with significant motion changes. By combining autoregressive generation with the progressive view sampling strategy, we can achieve optimal performance, significantly enhancing spatiotemporal consistency, as demonstrated in Fig. 10(c) and Fig. 11(c). We further conduct additional visualizations of ablation studies on the ***Refinement*** stage. As shown in Fig. 12, removing the refinement stage results in noticeable appearance drift. In contrast, including this refinement significantly improves the spatial-temporal consistency of the 4D objects generated.

## E    MORE VISUALIZATIONS OF COMPARISONS WITH STATE-OF-THE-ART METHODS

In this section, we present additional detailed visual comparisons between our proposed method and other state-of-the-art approaches. As demonstrated in Fig. 13, Fig. 14 and Fig. 15, Consistent4D (Jiang et al., 2024b) tends to produce over-saturated outputs due to the limitations of SDS, while SV4D (Xie et al., 2024) results in overly blurred outputs due to domain gap issues. By integrating the ideas of Consistent4D and SV4D, where anchor multi-view sequences are first generated

through a multi-view diffusion model followed by SDS-based refinement, STAG4D (Zeng et al., 2025) achieves improved results. However, it still exhibits noticeable noise and unrealistic patterns. Moreover, due to limitations in the training datasets, both SV4D and STAG4D struggle to generate 4D objects from longer input videos, hindering their practical applications. In comparison, our proposed AR4D achieves clearer renderings, enhanced spatial-temporal consistency, and improved alignment with input prompts.

## F   MORE VISUALIZATIONS OF 4D ASSETS GENERATED BY AR4D

In this section, we provide more results of the 4D assets generated by our proposed AR4D. As demonstrated in Fig. 16, Fig. 17, Fig. 18, Fig. 19, Fig. 20, and Fig. 21, the rendered novel-view videos exhibit superior spatial-temporal consistency.

## G   LIMITATIONS AND FUTURE WORKS

Although our method adopts an autoregressive generation paradigm with the potential to support real-time streaming applications, it currently falls short of real-time performance. Specifically, the optimization for each frame takes approximately 30–40 seconds on a single A100 GPU, primarily due to the computational overhead introduced by the pre-trained expert models used in our pipeline. We acknowledge this as a practical limitation, and in future work, we aim to improve the efficiency of the underlying models and optimization strategies to move closer toward real-time deployment. Additionally, while our method is SDS-free and achieves strong performance on complex scenes, it remains constrained by the limitations of the pre-trained large-scale 3D reconstruction models. We plan to address this by developing stronger reconstruction priors, such as incorporating optical flow or dynamic scene understanding modules. Regarding potential societal impact, our work is primarily intended for applications such as virtual reality, digital humans, and immersive content creation. Nevertheless, as with many generative technologies, we acknowledge the potential risks of misuse in the creation of synthetic content or deepfakes. To mitigate this, we encourage responsible usage and the incorporation of content authentication and detection systems in downstream applications.

## THE USE OF LARGE LANGUAGE MODELS (LLMS)

Large language models (LLMs) were employed solely for grammar correction and minor improvements in readability. They were not involved in any other aspects of the research process.

## ETHICS STATEMENT

This work does not involve human participants, animal studies, or the use of personally identifiable or sensitive data. The research does not pose foreseeable risks related to harm, bias, discrimination, misuse, or ethical concerns regarding privacy, security, or compliance. The authors declare no conflicts of interest or external sponsorship that could have influenced the reported results.

## REPRODUCIBILITY STATEMENT

We have made every effort to support reproducibility. Detailed descriptions of experimental settings, hyperparameters, and implementation choices are provided in the main text and appendix. The complete source code will be released upon publication to enable independent verification and facilitate further research.

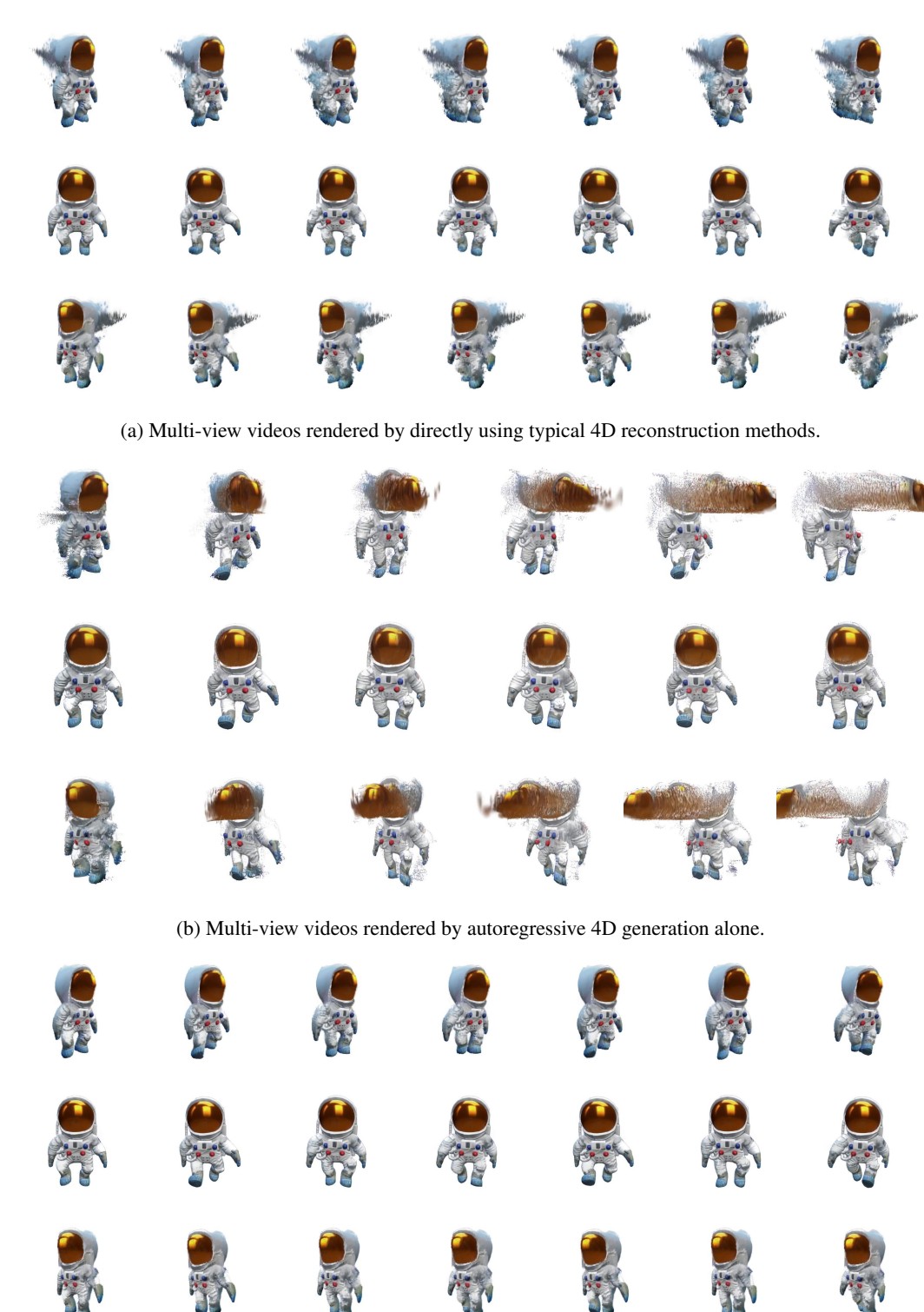

(a) Multi-view videos rendered by directly using typical 4D reconstruction methods.

(b) Multi-view videos rendered by autoregressive 4D generation alone.

(c) Multi-view videos rendered by autoregressive 4D generation with the progressive view sampling strategy.

Figure 10: Additional visualizations from the ablation studies on integrating autoregressive 4D generation and progressive view sampling strategy.

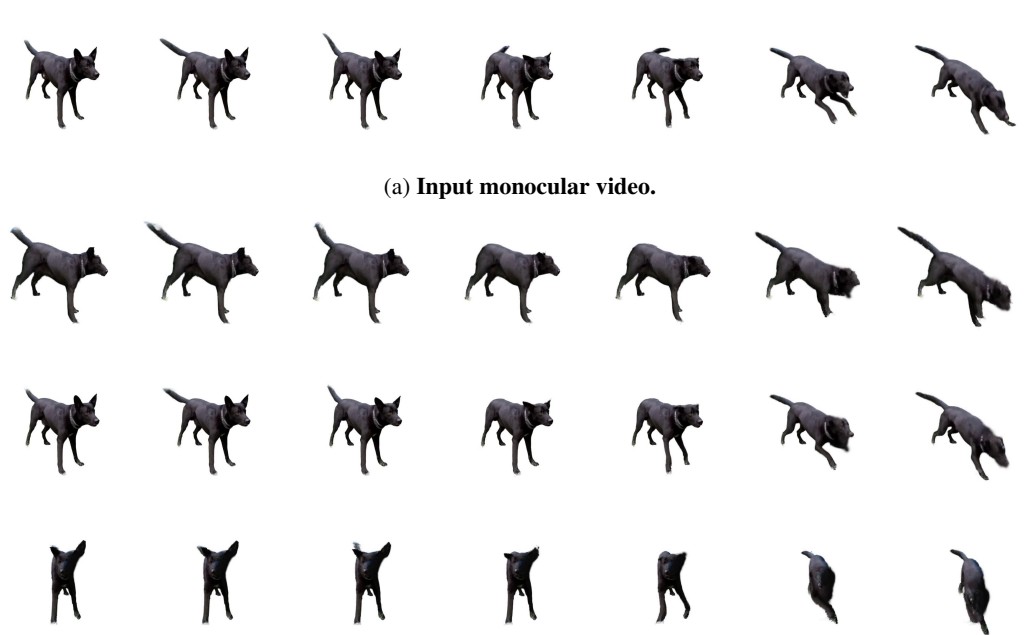

(a) **Input monocular video.**

(b) **Rendered multi-view videos without autoregressive generation:** precise motion estimation is challenging, especially for frames with substantial motion changes.

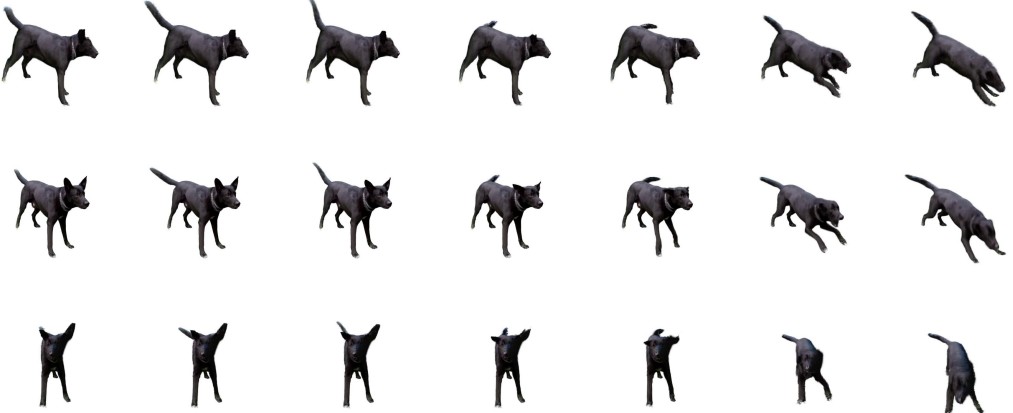

(c) **Rendered multi-view videos with autoregressive generation:** incorporating autoregressive generation enhances motion and geometry estimation, leading to more accurate and consistent results.

Figure 11: More visualizations of ablation studies on whether incorporating autoregressive generation.

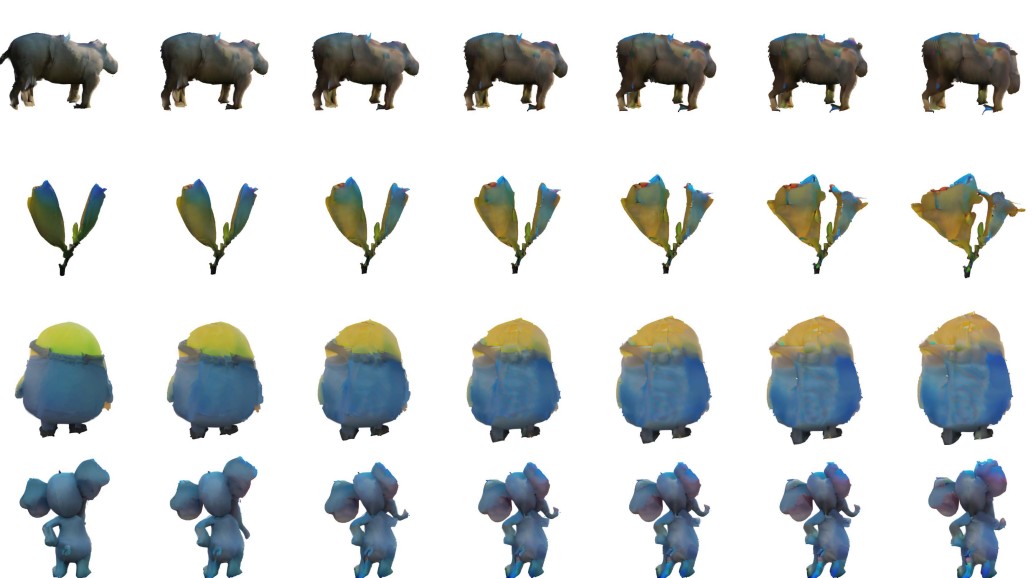

(a) Results obtained without the refinement stage, obvious appearance drift can be observed.

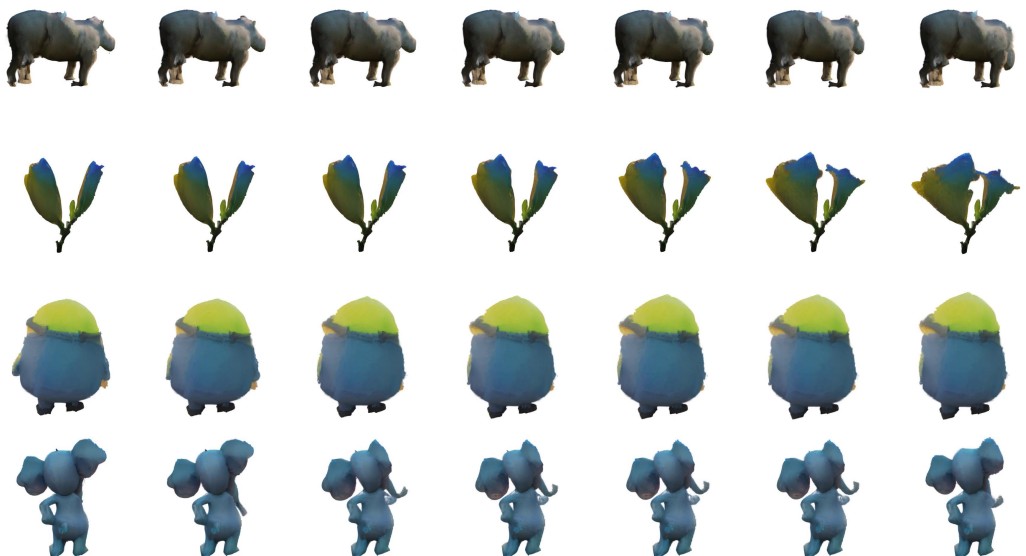

(b) With the refinement stage, appearance drift can be addressed, leading to results with better spatial-temporal consistency.

Figure 12: More visualizations of ablation studies on whether incorporating the refinement stage.

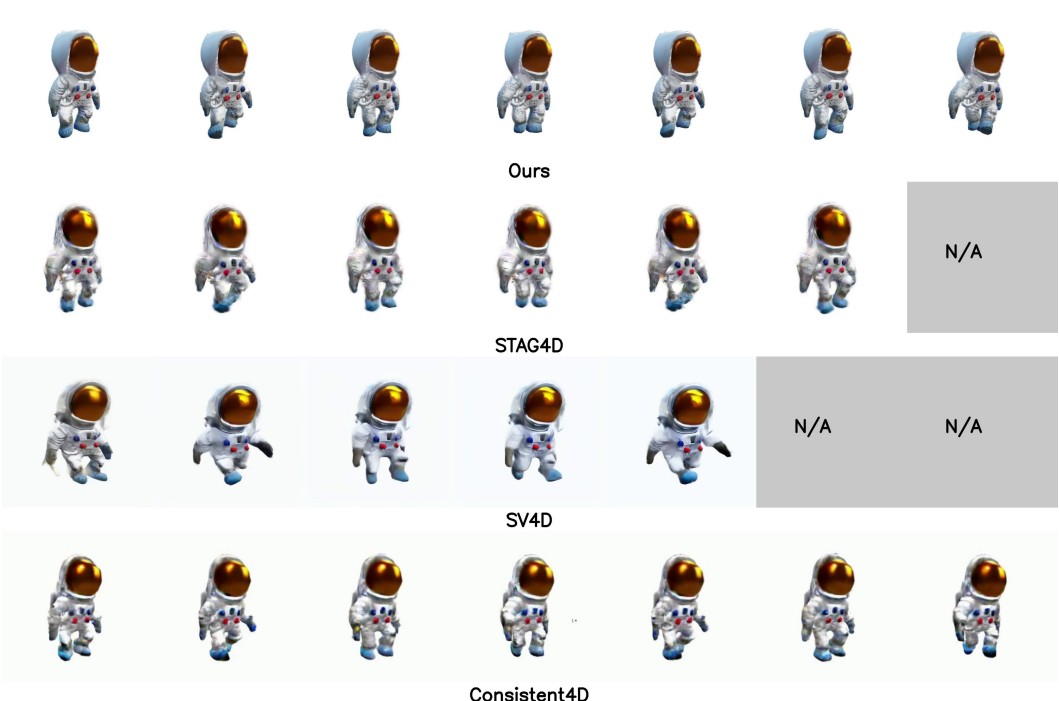

(a) Comparison of novel-view videos rendered by our method and other state-of-the-art methods at novel view 1.

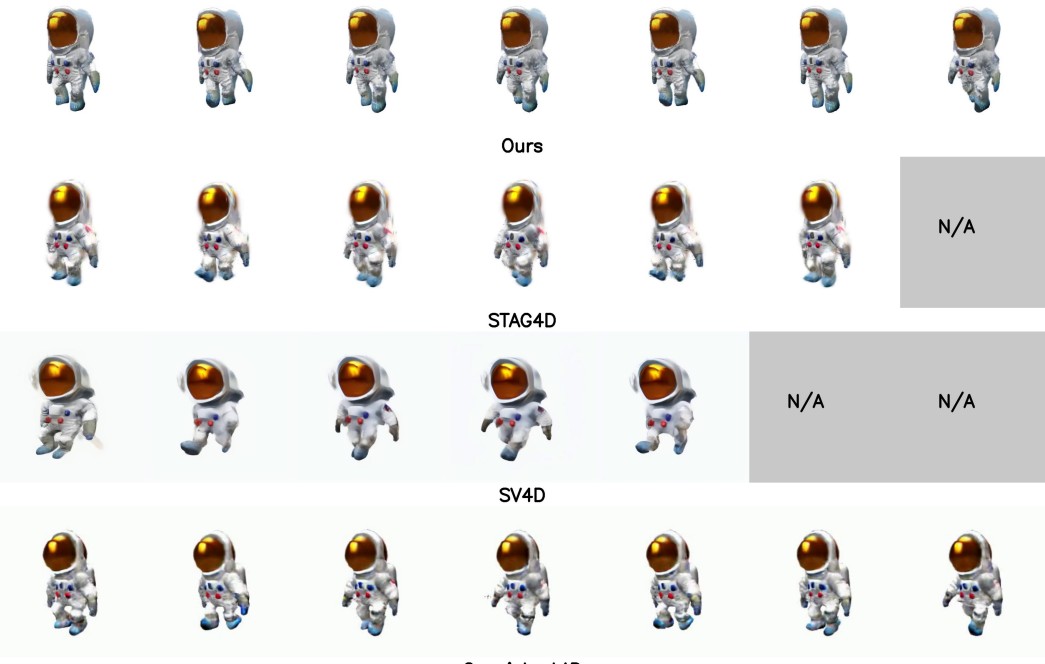

(b) Comparison of novel-view videos rendered by our method and other state-of-the-art methods at novel view 2.

Figure 13: More visualizations of comparison of novel-view videos rendered by our method and other state-of-the-art methods at different novel views on the task of Video-to-4D. *N/A* indicates that the corresponding method fails to generate novel views for the current frame.

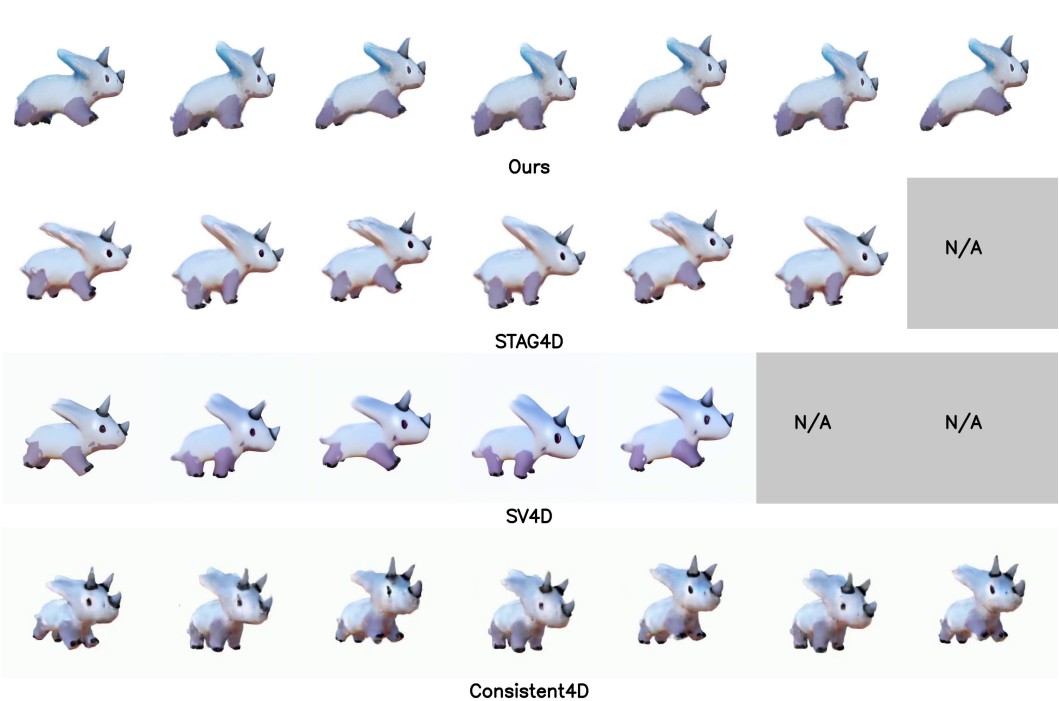

(a) Comparison of novel-view videos rendered by our method and other state-of-the-art methods at novel view 1.

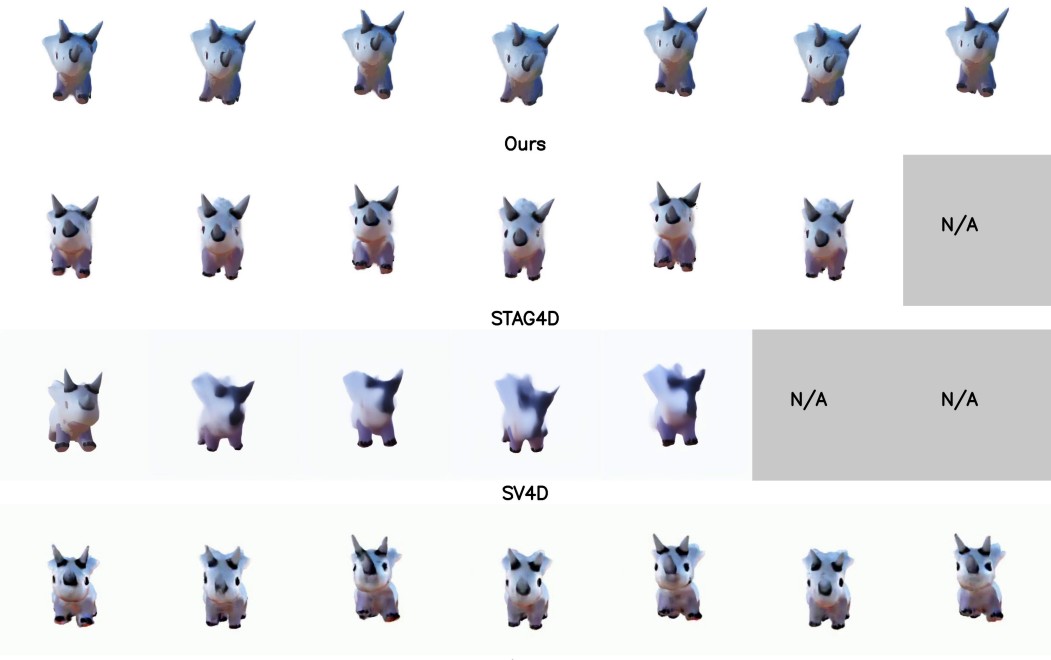

(b) Comparison of novel-view videos rendered by our method and other state-of-the-art methods at novel view 2.

Figure 14: More visualizations of comparison of novel-view videos rendered by our method and other state-of-the-art methods at different novel views on the task of Video-to-4D. *N/A* indicates that the corresponding method fails to generate novel views for the current frame.

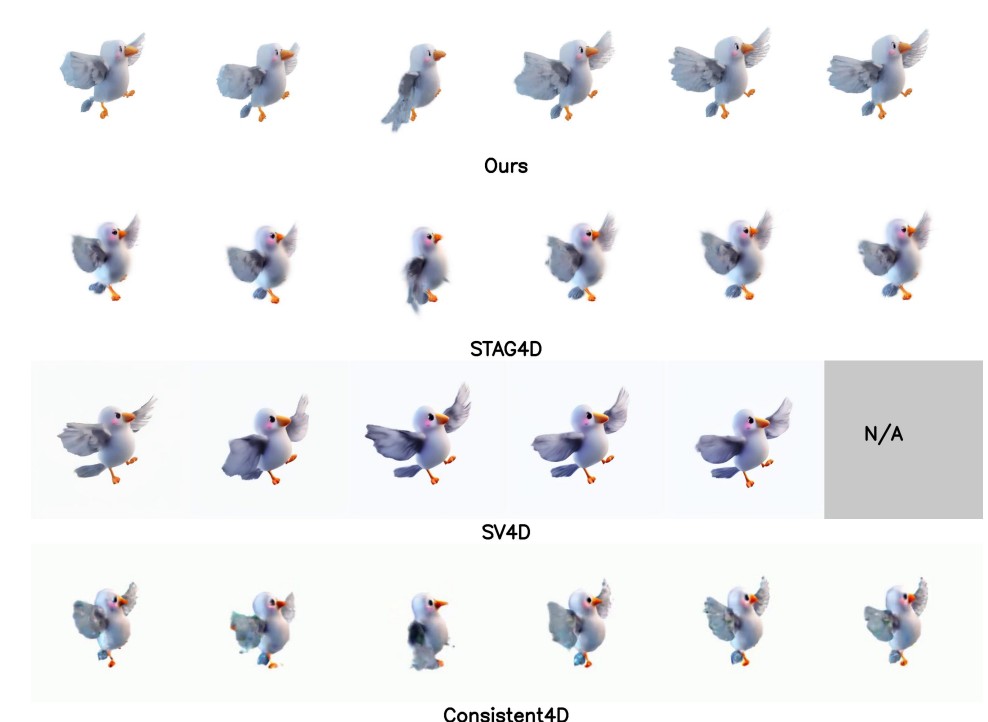

(a) Comparison of novel-view videos rendered by our method and other state-of-the-art methods at novel view 1.

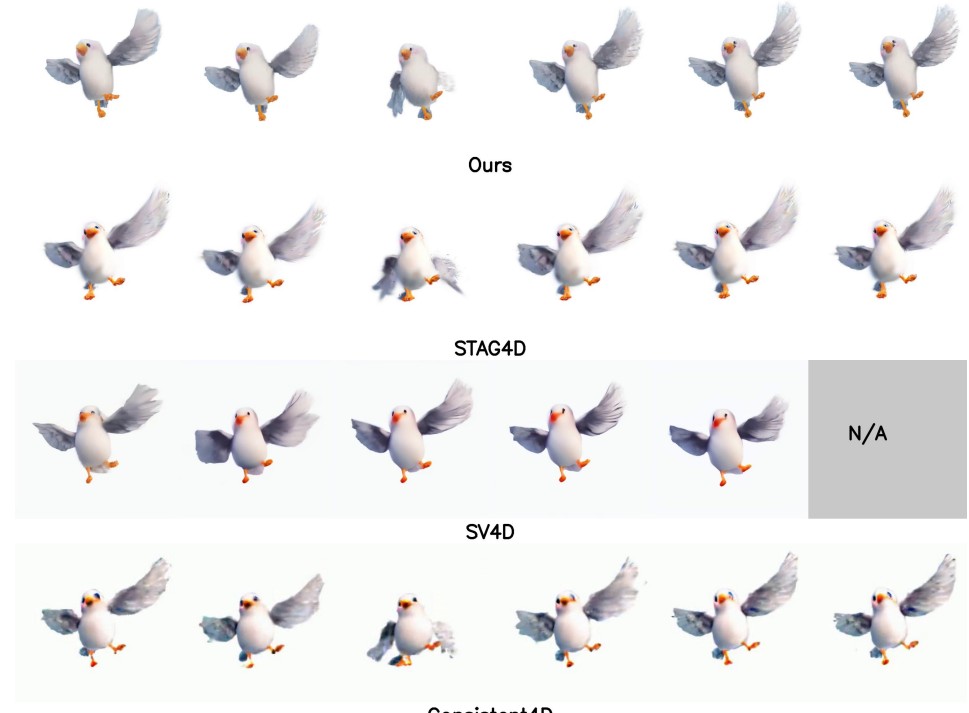

(b) Comparison of novel-view videos rendered by our method and other state-of-the-art methods at novel view 2.

Figure 15: More visualizations of comparison of novel-view videos rendered by our method and other state-of-the-art methods at different novel views on the task of Text-to-4D. *N/A* indicates that the corresponding method fails to generate novel views for the current frame.

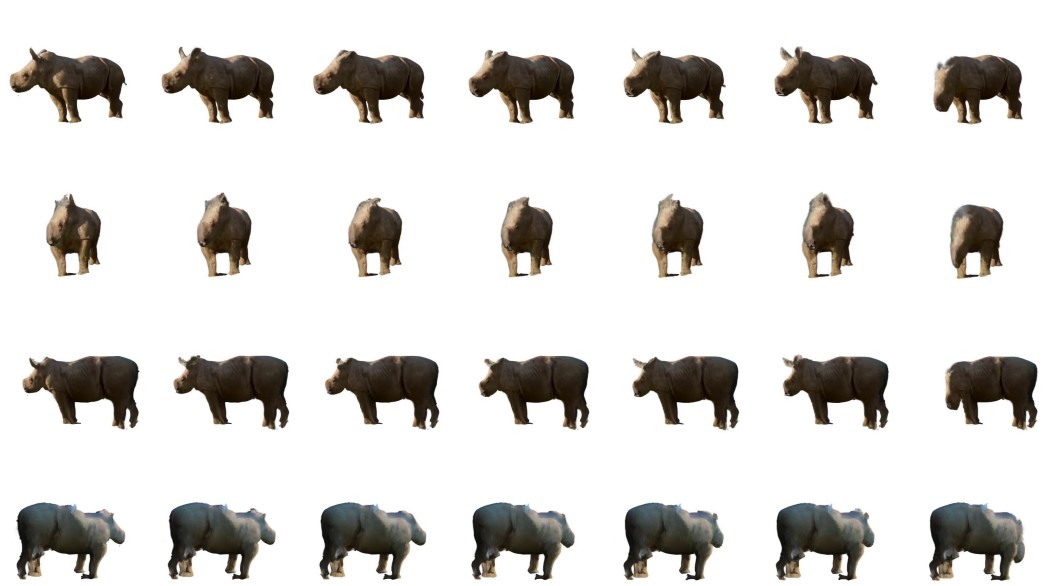

Figure 16: Additional results of multi-view videos rendered by AR4D, with the azimuth angles of $0°, -45°, 45°, 180°$ respectively.

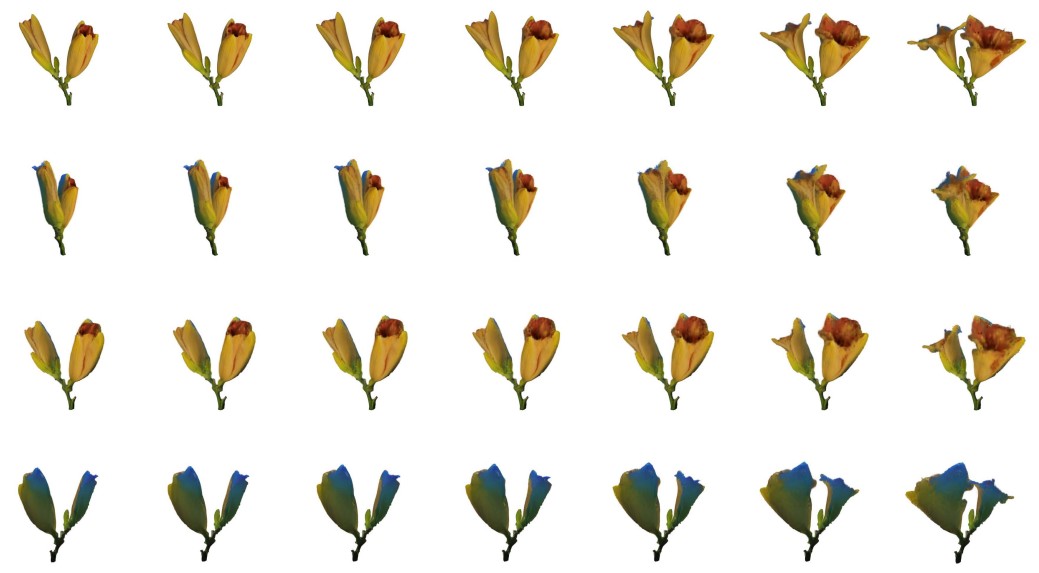

Figure 17: Additional results of multi-view videos rendered by AR4D, with the azimuth angles of $0°, -45°, 45°, 180°$ respectively.

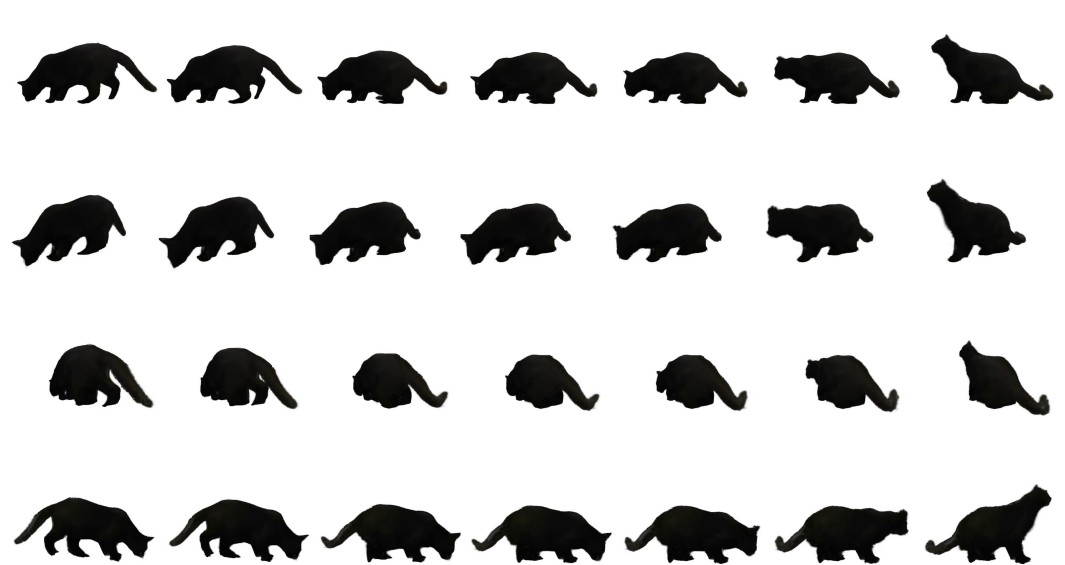

Figure 18: Additional results of multi-view videos rendered by AR4D, with the azimuth angles of $0°, -45°, 45°, 180°$ respectively.

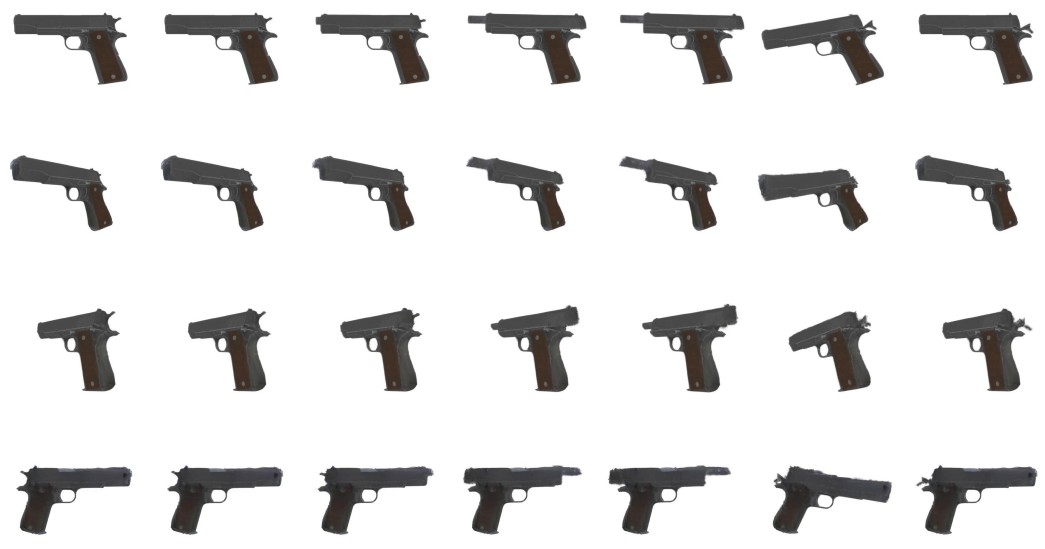

Figure 19: Additional results of multi-view videos rendered by AR4D, with the azimuth angles of $0°, -45°, 45°, 180°$ respectively.

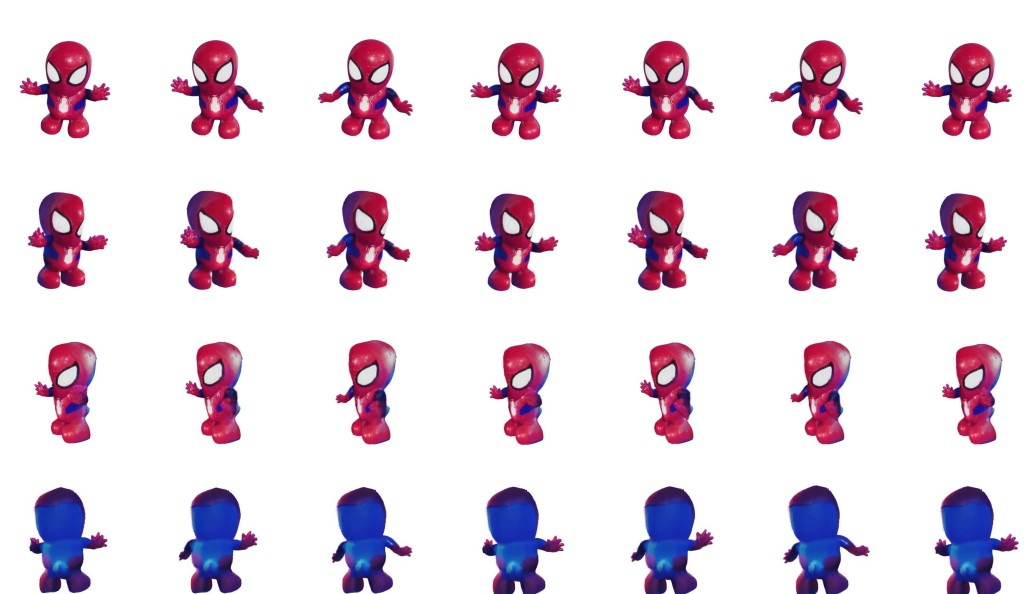

Figure 20: Additional results of multi-view videos rendered by AR4D, with the azimuth angles of $0°, -45°, 45°, 180°$ respectively.

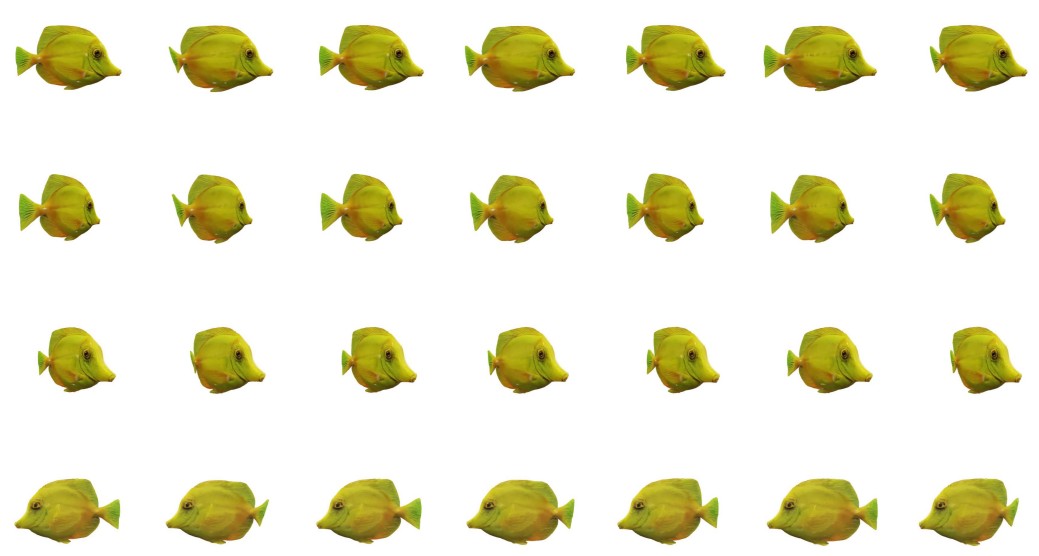

Figure 21: Additional results of multi-view videos rendered by AR4D, with the azimuth angles of $0°, -45°, 45°, 180°$ respectively.

