# OpenReview forum: "AR4D: Autoregressive 4D Generation from Monocular Videos"
_ICLR.cc/2026/Conference — ICLR 2026 Conference Withdrawn Submission_

### Official Review · Reviewer_gyGG · 2025-10-30

**Soundness:** 3
**Presentation:** 3
**Contribution:** 3
**Rating:** 6
**Confidence:** 4

**Summary:**

This paper proposes an autoregressive 4D generation pipeline from a monocular video without the time-consuming SDS loss.

The pipeline consists of three stages: 1. It utilizes the LGM to create a 3D representation of the first frame, then 2. It generates each frame’s 3D representation based on its previous frame’s representation. A progressive view sampling strategy is proposed to utilize priors from pretrained large-scale 3D reconstruction models. 3. A refinement stage is incorporated to avoid appearance drift.

Experiments demonstrate the state-of-the-art performance and faster generation speed.

**Strengths:**

1. The paper is well-written and easy to follow.
2. The proposed autoregressive generation framework, which leverages a pretrained large reconstruction model followed by global refinement, is both interesting and novel.
3. The extensive experimental results and ablation studies convincingly demonstrate the effectiveness of the proposed submodules and the superior performance of the overall approach.

**Weaknesses:**

1. As a 4D generation task, videos rendered across different time steps and views are crucial for evaluating the fidelity and consistency of the generated 4D objects. However, I could not find the rendered videos in the supplementary materials or on any linked website. If such videos are available, it would be helpful for the authors to provide a clear pointer to them in the rebuttal.

2. The paper currently reports only the overall time cost. A more detailed runtime analysis, breaking down the computational time for each module within the proposed framework, would be valuable for understanding efficiency and identifying potential bottlenecks.

**Questions:**

Theoretically, the proposed 4D generation framework should be applicable to longer input videos. Have the authors conducted any experiments or provided results demonstrating the method’s performance on longer sequences? Including such results would help illustrate the scalability and robustness of the approach.

---

### Official Review · Reviewer_8Tuf · 2025-10-30

**Soundness:** 4
**Presentation:** 2
**Contribution:** 2
**Rating:** 4
**Confidence:** 4

**Summary:**

The paper presents an autoregressive model for 4D generation from a monocular video. Previous works primarily rely on SDS loss to optimize 3D models, which often produces noticeable artifacts and is time-consuming. This work proposes an SDS-free, three-stage framework for 4D content generation. First, an off-the-shelf 3D generator is used for initialization. Next, an autoregressive model sequentially generates the 4D content, and a subsequent refinement network improves overall quality. Experiments show the proposed method achieves state-of-the-art results.

**Strengths:**

The paper is well written and easy to follow.

The progressive view-sampling strategy is efficient and provides useful insights.

The paper contains no obvious technical flaws.

**Weaknesses:**

- Although the authors claim SDS-free 4D generation, they state that MVDream was used for initialization; MVDream is an SDS-based optimization method.

- The overall framework feels incremental and the three-stage pipeline is complex and heavy, which raises concerns about reproducibility. Will the authors release code and sufficient implementation details? In addition, the baselines are rather old (e.g., SV4D, Consistent4D); please compare with more recent methods such as CAT4D, DreamMesh4D, In-2-4D, FB-4D, and Video4DGen.

- The generated 4D results do not show a clear qualitative improvement over prior work; I do not observe an obvious visual margin compared to earlier methods.

- The quantitative evaluation relies mainly on 2D metrics (PSNR, LPIPS, etc.). For 4D generation, please include temporal / 4D consistency metrics (e.g., multi-view or temporal consistency measures) instead of depending solely on frame-level scores.

- What do the authors mean by “dynamic visualization results”? For 4D generation, please provide dynamic video renderings (continuous sequences), not only static images. I could not find such videos in the supplementary material.

**Questions:**

What is the inference time for a  4D generation process? For example, MVDream reportedly requires about 15 minutes of optimization—please specify whether this is comparable to your method. Provide a detailed breakdown of runtime for each of the three stages (preferably in a table).

Missing some relevant references that should be considered for inclusion:
- CAT4D: Create Anything in 4D with Multi-View Video Diffusion Models
- In-2-4D: Inbetweening from Two Single-View Images to 4D Generation
- FB-4D: Spatial–Temporal Coherent Dynamic 3D Content Generation with Feature Banks
- Video4DGen: Enhancing Video and 4D Generation through Mutual Optimization
- Sweetdreamer: Aligning Geometric Priors in 2D Diffusion for Consistent Text-to-3D
 - Richdreamer: A Generalizable Normal–Depth Diffusion Model for Detail Richness in Text-to-3D

Please add these works to the references and, where relevant, include comparison experiments or discussions.

As noted in the Strengths and Weaknesses, I am currently inclined to give a borderline-reject score; I would be willing to raise this if the authors address the timing, baselines, and other main concerns.

---

### Official Review · Reviewer_4RqS · 2025-10-31

**Soundness:** 2
**Presentation:** 2
**Contribution:** 2
**Rating:** 4
**Confidence:** 4

**Summary:**

The authors propose to generation 4D object from monocular video in auto-regressove manner. Their framework consists of three stages:
initialization, generation (auto-regressive), refinement (global optimization). During generation, a MLP is optimized for adjacent veiws as local deformation field. Besides reconstruction loss of refernece frames, they leverage the renderings of per-frame generated 3D object as pesudo grouth truths in novel views. They carefully design a progressive view sampling strategy for the novel view training. At last, the optimize a global deformation field. Extensive experiments on public and private dataset demonstrate the effectiveness of the proposed method

**Strengths:**

1) the paper is well-organized and easy to follow

2) Experimental results indicate the model achieve state-of-the-art performance

3) The auto-regressive manner is novel in 4D generation

**Weaknesses:**

1) The authors did not clearly explain why the auto-regressive manner leads to accurate motion and geometry in 4D generation. Is it because only minor deformations are learned when optimizing on just two views, and the test data happen to exhibit only small deformations between adjacent views? Without the auto-regressive manner, does the learned global deformation field tend to introduce larger deformations between adjacent views?
2) The authors use per-frame LGM results as pseudo ground truth for novel views without incorporating any additional supervision signals. However, LGM results are inevitably temporally inconsistent. I wonder how the authors address this temporal inconsistency.
3) The proposed method introduces auto-regressive generation, yet later requires a global deformation field optimization during the refinement stage, which appears somewhat contradictory.
4) The proposed method introduces auto-regressive generation, yet later requires a global deformation field optimization during the refinement stage, which appears somewhat contradictory.As a 4D generation work，the authors are encouraged to provide video (both mulit-view and oribit) as supplementary material for better understanding of the performance of this work.

**Questions:**

See weakness. Besides, the authors are encouraged to use Multi-view video generation model results instead of LGM results as pesudo views to enhance the results. (I wonder why the authors didn't use those models)

---

### Official Review · Reviewer_wgAr · 2025-10-31

**Soundness:** 2
**Presentation:** 2
**Contribution:** 2
**Rating:** 4
**Confidence:** 4

**Summary:**

The paper proposes AR4D, a three-stage, SDS-free pipeline for lifting a monocular video to a dynamic 3D Gaussian (4D) sequence:

1. Initialization: generate orthogonal views of the first frame (e.g., via MVDream), reconstruct a 3D Gaussian, then fine-tune only positions/colors to better match the reference while keeping geometry fixed as the canonical space.

2. Generation: autoregressive per-frame local deformation fields (one MLP per adjacent frame pair) instead of a global field, trained against the next video frame.

3. Progressive view sampling (PVS): periodically render orthogonal views of the current estimate, reconstruct pseudo 3D with LGM, and impose RGB+depth consistency from gradually widening azimuth ranges to curb overfitting.

4. Refinement: a global deformation field from the canonical frame, constrained by per-frame depth to reduce appearance drift while preserving geometry.

Experiments follow STAG4D protocols across ~50 scenes for text-to-4D and video-to-4D, and the Consistent4D 8-scene benchmark, reporting PSNR/SSIM/LPIPS on chosen views plus CLIP-S and FVD, showing favorable results over baselines.

**Strengths:**

- The method does not require SDS, which is an interesting departure from other methods. Since SDS promotes mode-seeking, the generated results of SDS-free methods often contain more detail.

- Extensive experiments and visualizations demonstrate the performance improvements over baselines.

- Ablations clearly show the effects of each proposed component on the final performance.

**Weaknesses:**

- No user study or pairwise preference test is presented. For a generative paper, even a small-scale blind study would substantiate perceptual claims beyond LPIPS/FVD/CLIP-S.

- It seems that this method would not work on multi-objects due to the choice of pretrained priors. Limitations note dependence on current reconstructors; multi-object robustness isn’t demonstrated. Add experiments on multi-object scenes or state this as a limitation.

- The paper repeatedly attributes SDS methods’ limited diversity to SDS randomness, and claims AR4D offers "greater diversity", but the pipeline appears largely deterministic once the first-frame 3D is fixed. Appendix A even suggests retrying MVDream until satisfactory, which is not a controlled diversity mechanism.

- No hyperparameter sensitivity or robustness analyses for PVS.

- There are many recent papers relevant to this task like Vidu4D, SV4D, 4Diffusion, STP4D, Splat4D. A brief discussion or comparison with these methods is greatly preferred.

**Questions:**

Please see the weakness section. In addition, I have the following question:

- I see that the authors report the peak VRAM (30–40GB) usage; however, the memory vs. number of frames scaling is not shown. I am curious about the scalability of this method to longer videos.

---

### Note · Authors · 2025-11-13

I have read and agree with the venue's withdrawal policy on behalf of myself and my co-authors.